# Subsistence, Environment, and Society in the Taihu Lake Area during the Neolithic Era from a Dietary Perspective

**Yingying Wu [1], Can Wang [1,\*], Zhaoyang Zhang [1] and Yong Ge [2]**

[1] Department of Archaeology, School of History and Culture, Shandong University, Jinan 250100, China; 202111736@mail.sdu.edu.cn (Y.W.); 202111739@mail.sdu.edu.cn (Z.Z.)

[2] Department of Archaeology and Anthropology, University of Chinese Academy of Sciences, Beijing 100049, China; yongge@ucas.ac.cn

\* Correspondence: shandawangcan@sdu.edu.cn

**Abstract:** The Taihu Lake region is an important area where China's rice agriculture originated and where early Chinese civilisation formed. Knowing how this ecologically sensitive area's Neolithic residents adapted to environmental changes and utilised natural resources is key to understanding the origins of their agricultural practices and civilisation. Focusing on food resources, we systematically organised data from archaeobotanical and zooarchaeological research, human bone stable isotopic analyses, and fatty acid and proteome residue analyses on the Taihu Lake area's Neolithic findings to explore the interrelationships between subsistence, the environment, and society through qualitative and quantitative analysis supported by paleoenvironmental and archaeological evidence. The results showed that during the Neolithic era (7.0–4.3 ka BP), under a suitable climate with stable freshwater wetland environments, 38 varieties of edible animals and plants were available to humans in the Taihu Lake area. Despite agriculture being an important food source, rice cultivation and husbandry developed at different paces. Paddy rice cultivation began in wetlands and had always dominated the subsistence economy, as although gathering was universal and diverse, it produced a relatively low volume of food. In contrast, husbandry did not provide sufficient meat throughout the 2000 years of the Majiabang and Songze Cultures. Thus, fishing for freshwater organisms and hunting for wild mammals were the main meat sources before the domestication of pigs became the primary source of meat during the Liangzhu Cultural period. With the available wetland ecological resources and paddy rice farming (the sole crop), the Taihu Lake area transformed into an agricultural society in which rice cultivation dominated the Songze Culture's subsistence economy, which was also the first to exhibit social complexity. Then, finally, early civilisation developed in the Liangzhu Cultural period. This study contributes to understanding the unique evolutionary path of early Chinese civilisation and has important implications on sustainable resource utilisation for constructing ecological civilisations in present-day societies.

**Keywords:** Taihu Lake area; subsistence economy; wetland environment; rice agriculture; early civilisation

## 1. Introduction

"Food is the first necessity of the people" [1]. Food is the material basis of human evolution, the prime moving force in human societal development, and was the fundamental requirement for the formation of human civilisations [2,3]. The evolution of food resources also provides a record of the entire process of how humans have adapted to and transformed the external environment [4], thus revealing that ancient peoples' food types and structures have had a significant bearing on unearthing the history of their societies. Currently, archaeobotany, zooarchaeology, the stable isotopic analysis of human bones, and the residue analysis of artefacts are used together as the main method for comprehensively understanding the food resource usage of ancient people. This combination can also

provide new evidence and perspectives that further reveal the interactions between ancient peoples and their environment, and reconstruct the development path of subsistence economies, the spread of agriculture, and the growth of social complexity [5–8].

The region around Taihu Lake, which covers the South Jiangsu, North Zhejiang, and Shanghai regions, is an important area where China's rice agriculture originated, and where early Chinese civilization formed. It is also an area with a relatively sensitive ecological environment that is frequently impacted by extreme climate and environmental events [9–14]. Previous studies have shown that during the Neolithic era, the area around Taihu Lake suffered from multiple climatic and environmental fluctuations, driving the adaptive evolution of the ancients' food structure and subsistence economy [15–23]. However, most recent studies on this region used a single method to analyse the diet and subsistence of people at single sites and did not conduct a comparative analysis from an integrated compilation of the whole region. Therefore, the evolutionary processes, spatial differences, and influencing factors of the diet and subsistence of the Neolithic people in the area around Taihu Lake remain unclear.

This study systematically organised the data from archaeobotanical and zooarchaeological research, stable isotopic analyses of human bones, and residue analyses of the findings of the Majiabang, Songze, and Liangzhu cultures that existed around the Taihu Lake. This study also aimed to reveal the characteristics of the dietary and subsistence evolution processes and spatiotemporal variations of Neolithic people in the area through qualitative and quantitative analyses. Finally, the interrelationships between feeding behaviours towards animal and plant resources, the environmental changes, and the social development of ancient people have been discussed.

## 2. Study Area Overview

### 2.1. Natural Environment Overview

The area around Taihu Lake is located in the eastern part of the Yangtze River's lower valley, south of the Yangtze River estuary, north of the Hangzhou Bay and the Qiantang River, east of the Maoshan Mountain and the Tianmu Mountain, and west of the East China Sea (Figure 1). Taihu Lake is at the centre of the region, with mountains in the west and plains in the east, constituting a relatively independent and closed geographical unit (Figure 1). Contemporarily, the area around Taihu Lake has a subtropical monsoon climate, with mild and humid weather throughout the year and four distinct seasons. The mean annual temperature is 15–18 °C, and the mean annual precipitation is approximately 1300 mm. The typical zonal vegetation is evergreen broad-leaf forest and mixed evergreen and deciduous forest, and the major soil type is paddy soil [24]. Apart from a few low mountain and hilly areas, most of the plains in the area are <10 m above sea level. With abundant rainfall, the area has dense lakes and river networks, and transportation is convenient. The Jianghan area can be reached by travelling upstream along the Yangtze Valley, and the Haidai area and the Central Plains can be accessed by travelling northward through the Huanghuai Plain [25]. The favourable climate, topography, hydrological features, and soil conditions support rich animal and plant resources, which lay the foundation for the development of agriculture, human population growth, and the development of societies.

Numerous environmental changes occurred in the area around Taihu Lake during the late Pleistocene and Holocene periods. At the end of the Pleistocene period, during the early stages of land formation, the eastern plain had a dry and cold climate. The sea level was approximately −150 m, and most of the area was covered by terrestrial sediments with low organic matter content, on which weak meadow vegetation grew [26,27]. From approximately 10–7 ka BP, the climate changed from relatively dry and cold to warm and humid. From 8.2 ka BP, the area entered the Holocene Climate Optimum that was relatively stable, which allowed mixed evergreen and deciduous broad-leaved forests to develop. During this period, the largest transgression in the Holocene occurred [15,20,28–34]. At around 7 ka BP, the rise of the sea level slowed down, settling at slightly lower than the current level. The alluvium carried by the river accumulated to form land, and the Yangtze

River Delta began to accrete. At around 6–5.5 ka BP, the coastline to the east receded and reached south-eastern Zhejiang, although a period where sea levels eclipsed the current sea levels did not occur until 3 ka BP [20,30,35,36]. During the Majiabang Cultural period (7–5.8 ka BP), the major vegetation type was evergreen broad-leaf forests, and the climate was warm and humid. During the Songze Cultural period (5.8–5.3 ka BP), the proportion of evergreen broad-leaf trees decreased while the proportion of temperate deciduous broad-leaf trees and herbs increased, and the climate frequently fluctuated between warm–wet and warm–dry. During the Liangzhu Cultural period (5.3–4.3 ka BP), the major vegetation type was needle-leaf and broad-leaf mixed forests and grasslands dominated by needle-leaf trees, and the climate tended to be warm–dry and cool–dry. During the end of the Liangzhu Cultural period, events that involved significant temperature decreases, flooding, and transgression occurred [37–41].

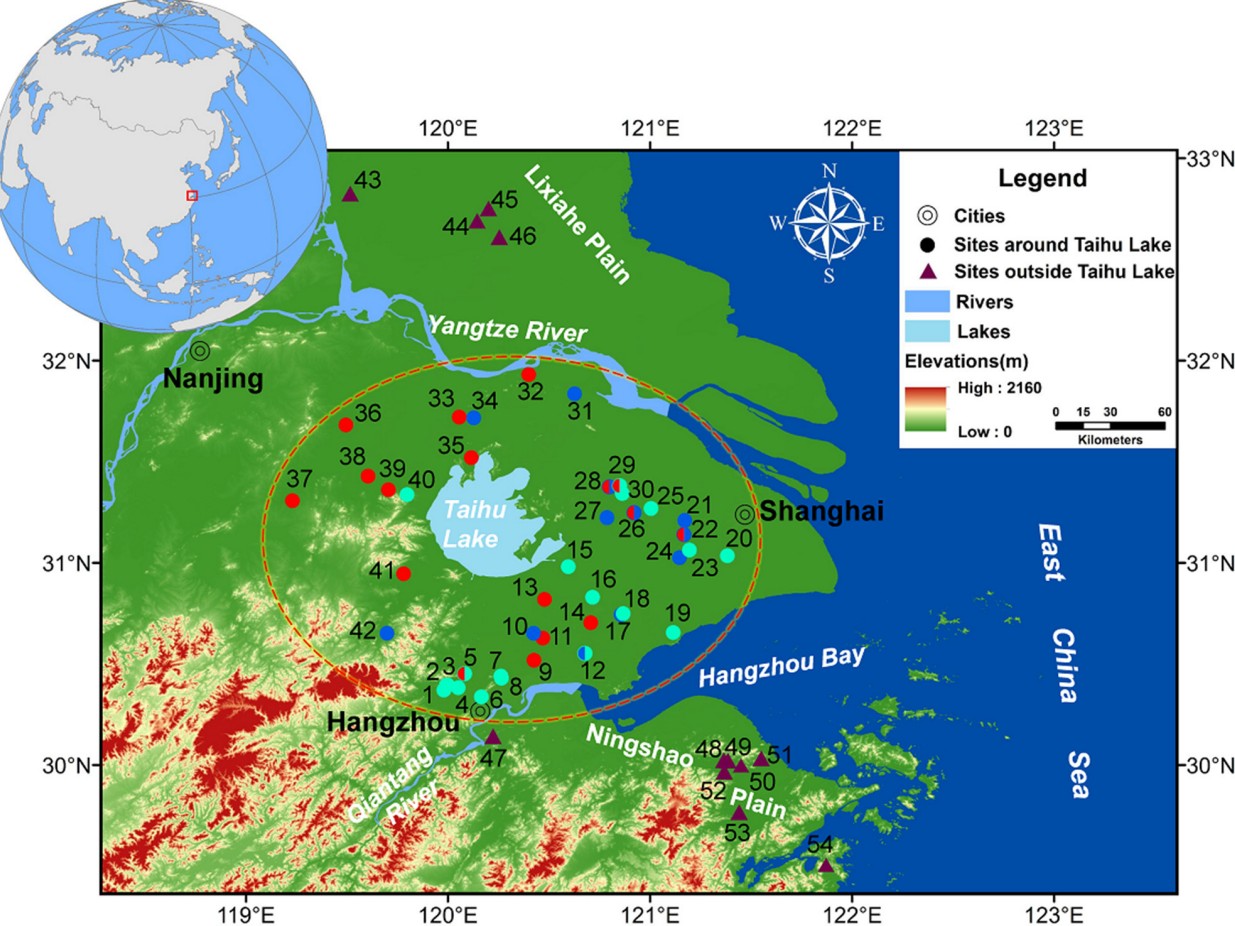

**Figure 1.** Location of the area around Taihu Lake and the distribution of Neolithic archaeological sites involved in this study: (1) Bianjiashan; (2) Mojiaoshan; (3) Meirendi; (4) Maoanli; (5) Nanzhuangqiao; (6) Shuitianfan; (7) Yujiashan; (8) Maoshan; (9) Xinqiao; (10) Dongjiaqiao; (11) Luojiajiao; (12) Xiaodouli; (13) Guangfucun; (14) Majiabang; (15) Longnan; (16) Shuangqiao; (17) Nanhebang; (18) Dafen; (19) Zhuangqiaofen; (20) Maqiao; (21) Fuquanshan; (22) Songze; (23) Guangfulin; (24) Yaojiaquan; (25) Shaoqingshan; (26) Jiangli; (27) Chenghu; (28) Caoxieshan; (29) Chuodun; (30) Zhumucun; (31) Xujiawan; (32) Dongshancun; (33) Xudun; (34) Qingchengdun; (35) Yangjia; (36) Sanxingcun; (37) Shendun; (38) Xixi; (39) Luotuodun; (40) Xiawan; (41) Jiangjiashan; (42) Yaodun; (43) Longqiuzhuang; (44) Kaizhuang; (45) Jiangzhuang; (46) Qingdun; (47) Kuahuqiao; (48) Jingtoushan; (49) Tianluoshan; (50) Cihu; (51) Yushan; (52) Hemudu; (53) Xiawangdu; (54) Tashan. Red dots—sites of Majiabang Culture; dark blue dots—sites of Songze Culture; light blue dots—sites of Liangzhu Culture; dots with two colors indicate that the site has two corresponding cultural attributes.

In the Neolithic era, the optimum climate, stable sea level, continuous expansion of land areas, and the formation of saucer-shaped depressions centred around Taihu Lake provided a broad space, wetlands, fresh water, and animal and plant resources—all of which were essential to the development of the Neolithic Cultures in around Taihu Lake. However, disastrous floods caused by the elaborate water system, sea-level fluctuations, and ocean storms that impacted the nearshore had certain levels of influence on the prehistoric environment, agriculture, and social formation of this area.

### 2.2. Overview of Cultures from an Archaeological Perspective

The area around Taihu Lake contained one of the most prosperous prehistoric cultures in China. The earliest Neolithic cultural remains are from the Luojiajiao site of the Majiabang Culture, dating back 7000 years to the late-Neolithic era, from which point a line of cultural succession of the Majiabang-Songze-Liangzhu Culture gradually formed [42,43].

The Majiabang Culture mainly appeared in the eastern part of the Taihu Lake area, during which time the Luotuodun Culture was more prominent in the western part. The Majiabang Culture can be divided into early and late periods, which spanned 7.0–6.5 ka BP and 6.5–5.8 ka BP, respectively, and experienced some overlap with the early and late Luotuodun Cultural periods. It is generally assumed that the Luotuodun Culture is a local cultural type of Majiabang Culture [44,45]. The sites of the Majiabang Culture cover tens of thousands to hundreds of thousands of square metres, and the Majiabang people adopted a broad-spectrum subsistence strategy—cultivating rice and raising livestock concurrently. The culture had a relatively strong cultural influence on the Ningshao Plain (Hemudu Culture), East JiangHuai (Longqiuzhuang Culture), and North Jiangsu (Beixin–Dawenkou Culture) regions. However, the sites of the Songze Culture are usually smaller, covering areas of thousands to tens of thousands of square metres. The agricultural techniques of this Culture improved significantly over a short time span, and agriculture played a relatively important role in their subsistence economy. The number of sites of the Liangzhu Culture increased drastically, with settlement clusters forming in various locations with an evident class polarisation. The settlement clusters shared a common identity: jade wares. Rice agriculture and husbandry were well developed, and the establishment of hydraulic systems in and around the ancient city of Liangzhu revealed the birth of an early state holding royal power over the region [42,43,46,47].

The archaeological data from the Taihu Lake area provides evidence of social stratification during the end of the Majiabang Culture and Songze Culture, making it the first area in China to undergo the social complexity process [43,48]. The abundant remains of rice and paddy field relics dating from the Majiabang to the Liangzhu Culture in the Taihu Lake area highlight the important role of rice cultivation in the origins of agriculture and the formation of paddy rice-farming traditions in East Asia [49–54].

## 3. Materials and Methods

Remains of animals and plants and their organic residues, as well as the stable isotopic signals from human tissues, are the most direct information sources for research on the diets of ancient people. This study used a systematic literature review—in which published data was collected from archaeobotanical and zooarchaeological research, stable isotopic analyses on human bones, and residue analyses (analysis of fatty acid and proteomics) from 42 Neolithic sites (Figure 1) in the Taihu Lake area (Table S1)—to extract information on animal and plant-based food resources. We also collected evidence of food resources from 12 sites (Figure 1) on the Ningshao and Lixiahe plains for comparative analysis and discussions (Table S2; Figure 1). All of the collected data were of provenances that had clear age ranges determined by radiocarbon dating and the features of cultural remains.

Before the data analysis, the existing studies were first categorised. Data not collected by "system sampling" (sampling with a scheme suitable for certain sites along with reasonable sample sizes and location) were from either a small sample ($n < 10$) or one with only categories but without quantitative information. These were classified as non-systematic

research data. The data classified as systematic research data were derived from studies with designed sampling strategies, large sample sizes, and that were published with comprehensive quantitative information. For the archaeological data of animals and plants obtained from non-systematic research (Tables S1(I) and S2(I)), and for the small amount of data from residue analysis, this study only conducted a qualitative analysis to investigate the variety and universality of food resources in different periods. The "universality" was the proportion of sites in which a certain taxon was found during certain periods. The number of taxa (NTAXA) and Simpson and Shannon–Wiener diversity indices were also used to measure the taxonomic richness and diversity of each period [55–57].

From the systematic studies on some sites, quantitative data such as the absolute counts, percentage, and ubiquity of plant remains, as well as the number of identifiable specimens (NISP) of animal remains (Table S1(II–IV)), were classified according to different cultural periods to conduct the comparative analysis of different food resources. The "percentage" data were obtained from original references and calculated for each site to show the proportion of taxa amongst the total number of identified taxa (Table S1(II–IV)). "Ubiquity" was a measure of how "commonly" some plants occurred in the sample elements, which was simply a proportion for each site: the number of samples in which certain plants were found divided by the total number of samples during certain periods (Table S1(IV)). In addition, the method of relative percentage was used to integrate the data of the number of animal (NISP) and plant (absolute counts) remains in the three periods to examine crops and the domestic animals as proportions of the diets of ancient peoples in different periods. Moreover, the variation trends were also examined, which reflect the temporal changes in human subsistence. The relative percentage analysis was calculated by $n_1/N_1 \times 100\%$, where $n_1$ is the number of each plant and animal group/type, and $N_1$ is the total number of all plant groups/types. Both are from selected sites for each cultural period (Table S1(II,III)). A total of 19 sites had available quantitative data for further analysis, including 6 Majiabang Culture sites, 3 Songze Culture sites, and 11 Liangzhu Culture sites (Table S1(II–IV)).

There are only seven sites in the area around Taihu Lake—the Sanxingcun, Majiabang, Xudun, Songze, Zhuangqiaofen, Jiangjiashan, and Meirendi sites—that provided stable isotopic data from human bones (Table 1). In addition, because the Ningshao Plain (located south of the Qiantang River) had a similar ecological environment to the Taihu Lake area, there had always been exchanges and interactions between the Hemudu Culture (7–5 ka BP) and the Neolithic Cultures in the area around Taihu Lake. Therefore, the stable isotopic data of human bones from three sites of the Hemudu Culture (namely the Tianluoshan, Hemudu, and Tashan sites) were selected for inclusion in the comparative analysis (Table 1). Thus, the proportion of $C_3$ and $C_4$ plants and meat in the people's diets at the examined sites were determined via the measured $\delta^{13}C$ and $\delta^{15}N$ analyses' values of the human bones.

Residue analysis has seldomly been used when studying the Taihu Lake area. A few cases of its application include lipid profiling to analyse the organic compounds adsorbed by clays and inner-wall carbonised materials of the pottery shreds from the Xiawan site belonging to the Liangzhu Culture in Yixing [58]. Its results (Table S1(I)) will be used in the following analysis and discussion.

**Table 1.** Human bone $\delta^{15}$N and $\delta^{13}$C values from the 10 sites used in this study.

| Site Names | Cultural Period | Sample Type | Mean $\delta^{13}$C Values (‰) | Mean $\delta^{15}$N Values (‰) | SD of $\delta^{13}$C Values (‰) | SD of $\delta^{15}$N Values (‰) | References |
|---|---|---|---|---|---|---|---|
| Tianluoshan (*n* = 10) | Hemudu Culture | Bone collagen | −20.7 | 8.7 | 0.5 | 0.9 | [59] |
| Hemudu (*n* = 2) | Hemudu Culture | Bone collagen | −16.7 | 11.4 | 2.3 | 0.3 | [60] |
| Tashan (*n* = 3) | Hemudu Culture | Bone collagen | −18.4 | 9.2 | 0.5 | 0.7 | [61] |
| Majiabang (*n* = 22) | Majiabang Culture | Bone bioapatite | −11.7 | —— | 0.4 | —— | [62] |
| Sanxingcun (*n* = 19) | Majiabang Culture | Bone collagen | −20.05 | 9.69 | 0.21 | 0.33 | [63] |
| Xudun (*n* = 2) | Majiabang Culture | Bone collagen | −20.2 | 10.4 | 0.3 | 0.5 | [64] |
| Jiangjiashan (*n* = 2) | Majiabang Culture | Bone collagen | −20.31 | 7.62 | 0.22 | 3.59 | [65] |
| Songze (*n* = 2) | Songze Culture | Bone collagen | −19.89 | 10.9 | 0.5 | 1.7 | [60] |
| Zhuangqiaofen (*n* = 22) | Liangzhu Culture | Bone bioapatite | −12.8 | —— | 0.7 | —— | [64] |
| Meirendi (*n* = 9) | Liangzhu Culture | Bone collagen | −19.9 | 10.3 | 1.3 | 0.4 | [66] |

## 4. Results

### 4.1. Edible Animal and Plant Resources in the Area around Taihu Lake during the Neolithic Era

During the Neolithic era, the ancient people living around the Taihu Lake area consumed an extremely wide variety of animals and plants (38 varieties; Table S1(I)). The plants consumed included four major categories: cereals, wild grasses, nuts, and fruits and vegetables. There were four varieties of cereals—rice (*Oryza sativa*), foxtail millet (*Setaria italica*), common millet (*Panicum miliaceum*), and soybeans (*Glycine* spp.); one variety of wild grasses—wild rice (*Oryza rufipogon*); three varieties of nuts—acorns (*Quercus* sp.), water caltrops (*Trapa* sp.), and semen euryales (*Euryale ferox*); and twelve varieties of fruits and vegetables—cucurbits (*Lagenaria siceraria*), melons (*Cucumis* sp.), lotus seeds (*Nelumbo nucifera*), persimmons (*Diospyros* sp.), jujubes (*Choerospondias axillaris*), grapes (*Vitis* sp.), peaches (*Prunus persica*), apricots (*Prunus armeniaca*), dark plums (*Prunus mume*), plums (*Prunus salicina*), cherries (*Cerasus* spp.), and hawthorns (*Crataegus* sp.). There were three categories of consumed meant: terrestrial animals, aquatic animals, and birds. All the varieties of terrestrial animals were wild except for domestic pigs (*Sus scrofa* f. *domesticus*) and dogs (*Canis lupus familiaris*), which the people raised. The wild animals included buffaloes (*Bubalus* spp.), deer (Cervidae), raccoon dogs (*Nyctereutes procyonoides*), Caprinae, Leporidae, boar (*Sus scrofa*), and elephants (*Elephas maximus*). The aquatic animals mainly included five varieties of freshwater animals, including fishes, molluscs (shellfish, mussels, and snails), reptiles (crocodiles), amphibians (turtles and tortoises), and crabs (*Brachyura* spp.), whereas the marine varieties included whales, oceanic fishes, and shellfishes. The bird varieties were waterfowls such as ducks (*Anas* spp.), wild goose (*Anser*, *Cygnus*), and gulls (Laridae). Table 2 shows the detailed taxa used as food in the Taihu Lake area during the Neolithic era. The NTAXA of edible plants and animals are 20 and 63, respectively, indicating high dietary taxonomic richness. The diversity values for animals were generally greater than those for plants throughout the Neolithic era (Table 3), indicating a higher species diversity in the residents' meat resources than that obtained through gathering activities.

Considering the popularity of the various food resources by the standard of universality > 30% (appearing in ≥13 sites), the most frequently consumed food types of the Neolithic people in the Taihu Lake area included rice, water caltrops, semen euryales, cucurbits, melons, domestic pigs, dogs, deer, buffaloes, and freshwater animals (fishes, molluscs, and amphibians). The prehistoric people living around Taihu Lake mainly fed on terrestrial and freshwater food resources and seldomly consumed marine animals. The relatively low $\delta^{15}$N values and the negative $\delta^{13}$C values also illustrate this point (Table 1; Figure 2).

**Table 2.** List and NTAXA of plant and animal foods from archaeobotanical and zooarchaeological remains during each period of Neolithic in the area around Taihu Lake.

| Culture | Edible Plant Taxon | | | | | Animal Taxon | | | | |
|---|---|---|---|---|---|---|---|---|---|---|
| | Cereals | Wild Grasses | Nuts | Fruits and Vegetables | Total | Land Mammals | Freshwater Animals | Birds | Marine Resources | Total |
| Majiabang | *Oryza sativa* *Setaria italica* *Glycine* spp. | *Oryza rufipogon* | *Quercus* sp. *Trapa* sp. *Euryale ferox* | *Lagenaria siceraria* *Cucumis* sp. *Nelumbo nucifera* *Diospyros* sp. *Vitis* sp. *Prunus persica* *Prunus armeniaca* *Prunus mume* *Prunus salicina* *Cerasus* spp. *Crataegus* sp. | | *Sus scrofa* f. *domesticus* *Sus scrofa* *Canis* *Arctonyx* *Cervus axis* *Cervus nippon* *Elaphurus* *Mantiacus* *Bubalus* *Hydropotes* Leporidae *Nyctereutes* *Elephas maximus* Sciuridae *Herpestes* | *Crocodylus* *Trionyx* *Pelochelys* Ranidae Serpentiformes *Geoclemys* *Channa* *Mylopharyngodon* *Carassius* *Hypophthalmichthys* Cyprinidae *Lateolabrax* *Silurus* *Pelteobagrus* *Acipenser* *Brachyura* Viviparidae *Anodonta* *Lamprotula* *Cuneopsis* *Arconaia* *Hyriopsis* *Unio* Corbiculidae *Bellamya* | *Anas* Phalacrocracidae *Corvus* Laridae *Anser* Gruidae | Cetacea *Galeocerdo cuvier* *Mugil* *Helicolenus* *Trachurus* | |
| NTAXA | 3 | 1 | 3 | 11 | 18 | 15 | 25 | 6 | 5 | 51 |
| Songze | *Oryza sativa* *Panicum miliaceum* *Glycine* spp. | | *Trapa* sp. *Euryale ferox* | *Lagenaria siceraria* *Cucumis* sp. *Diospyros* sp. *Vitis* sp. *Prunus persica* *Prunus mume* | | *Sus scrofa* f. *domesticus* *Canis* *Arctonyx* *Elaphurus* *Cervus nippon* *Bubalus* *Lutra* *Hydropotes* *Mantiacus* | Testudinidae Cyprinidae | | | |

**Table 2.** *Cont.*

| Culture | Edible Plant Taxon | | | | | Animal Taxon | | | | |
|---|---|---|---|---|---|---|---|---|---|---|
| | **Cereals** | **Wild Grasses** | **Nuts** | **Fruits and Vegetables** | **Total** | **Land Mammals** | **Freshwater Animals** | **Birds** | **Marine Resources** | **Total** |
| NTAXA | 3 | 0 | 2 | 6 | 11 | 9 | 2 | 0 | 0 | 11 |
| Liangzhu | *Oryza sativa* *Setaria italica* *Glycine* spp. | *Oryza rufipogon* | *Quercus* sp. *Trapa* sp. *Euryale ferox* | *Lagenaria siceraria* Cucumis sp. Diospyros sp. Choerospondias axillaris Vitis sp. Prunus persica Prunus armeniaca Prunus mume Prunus salicina Cerasus spp. | | *Sus scrofa* f. *domesticus* Sus scrofa Canis Bubalus Cervus nippon Caprinae Hydropotes | Testudinidae Trionyx Cyprinidae Sinotaia Bellamya Unio Acuticosta Arconaia Cuneopsis Lanceolaria Lamprotula Corbicula Viviparidae | *Anser* Cygnus Anas | Chondrichthyes Ostreidae Meretrix Cyclina | |
| NTAXA | 3 | 1 | 3 | 10 | 17 | 7 | 13 | 3 | 4 | 27 |
| Neolithic NTAXA | 4 | 1 | 3 | 12 | 20 | 17 | 30 | 7 | 9 | 63 |

**Table 3.** Simpson and Shannon–Wiener diversity values for plant and animal foods in the area around Taihu Lake during each period.

| Culture | Food Types | Shannon–Wiener Diversity Values | Simpson Diversity Values |
|---|---|---|---|
| Majiabang | Plants | 1.11 | 0.52 |
|  | Animals | 1.77 | 0.76 |
| Songze | Plants | 0.45 | 0.21 |
|  | Animals | / | / |
| Liangzhu | Plants | 0.48 | 0.19 |
|  | Animals | 0.92 | 0.34 |

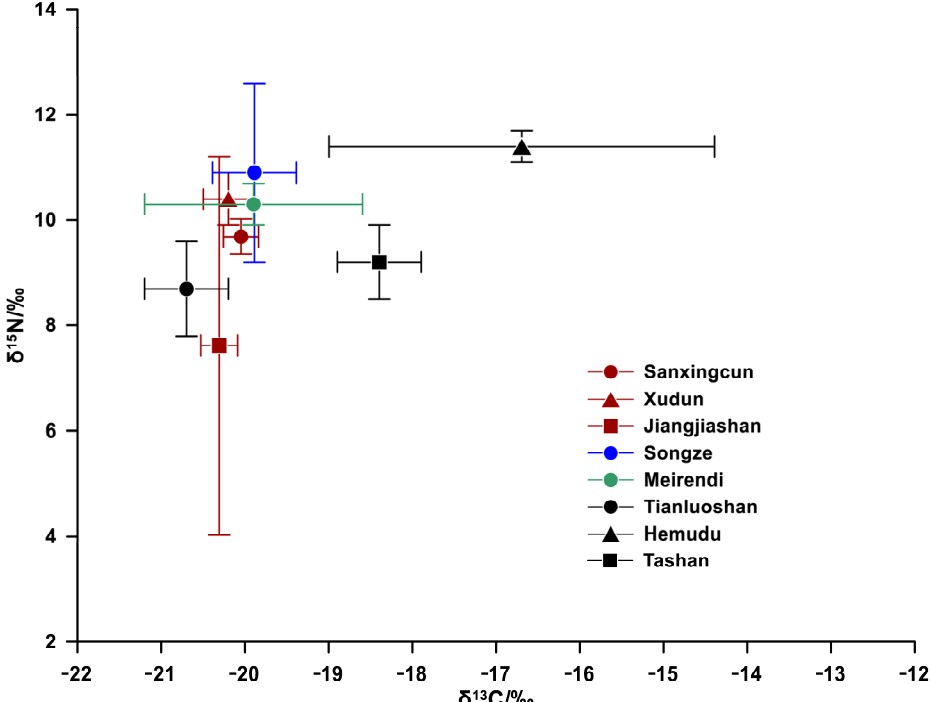

**Figure 2.** Scatter plot of $\delta^{13}$C and $\delta^{15}$N values of human bones from the area around Taihu Lake and Ningshao Plain. The red, blue, green, and black marks indicate the sites of the Majiabang, Songze, Liangzhu, and Hemudu Cultures, respectively. Majiabang and Zhuangqiaofen sites were not included in this figure because of the lack of $\delta^{15}$N values, which can be seen in Table 1.

### 4.2. Diet of Neolithic Ancient People Living around Taihu Lake in Different Periods and Temporal Changes

4.2.1. Majiabang Cultural Period (7.0–5.8 ka BP; 17 Sites)

Based on the total number of varieties, NTAXA, and the diversity values of the food resources (Table 2, Table 3 and Table S1(I)), the Neolithic people living around Taihu Lake were found to consume the highest richness of food during the Majiabang Cultural period; their food included 3 types of cereals (only common millet was not found), 2 types of domestic animals (pigs and dogs), 15 types of edible wild plants (only jujubes were not found), and 14 types of wild animals, which accounted for 90% of all the food resource types. The total NTAXA of food was 69, with the NTAXA of plants and animals being 18 and 51, respectively (Table 2). Among them, rice, pork, and dog meat were the most commonly found agricultural products, with universalities of 88%, 65%, and 71%, respectively. Foxtail millet only appeared at the Yangjia site in a small number (nine grains), and soybeans were only unearthed at the Dongshancun and Chuodun sites. Other commonly found food resources were water caltrops, semen euryales, cucurbits, deer, water buffaloes, birds, freshwater fishes, turtles, and molluscs. The food variety was wide by each site (mean of

9.5 types/site), especially in the Dongshancun site, where 25 types of food were consumed, including 4 types of marine fishes. Notably, lotus seeds, hawthorns, raccoon dogs, rabbits, elephants, crocodiles, whales, and most marine fishes did not appear in the diets of the Neolithic people living around Taihu Lake after the Majiabang Culture.

From the quantitative data (Figure 3), domesticated rice dominated the plant-based food resources of most of the sites (42–94%; relative percentage of 67%). Nuts, fruits, vegetables, and other edible wild plants accounted for a slightly smaller proportion (2.8–51%; relative percentage of 30%). Freshwater animals (27–80%; relative percentage of 60%) and terrestrial mammals (17–73%; relative percentage of 39%) were the major meat sources. Among mammals, domesticated pigs and dogs accounted for <23% of the consumed food in most sites, and pigs (11–22%; relative percentage of 93%) were generally consumed at a much higher proportion than dogs (0–23%; relative percentage of 7%).

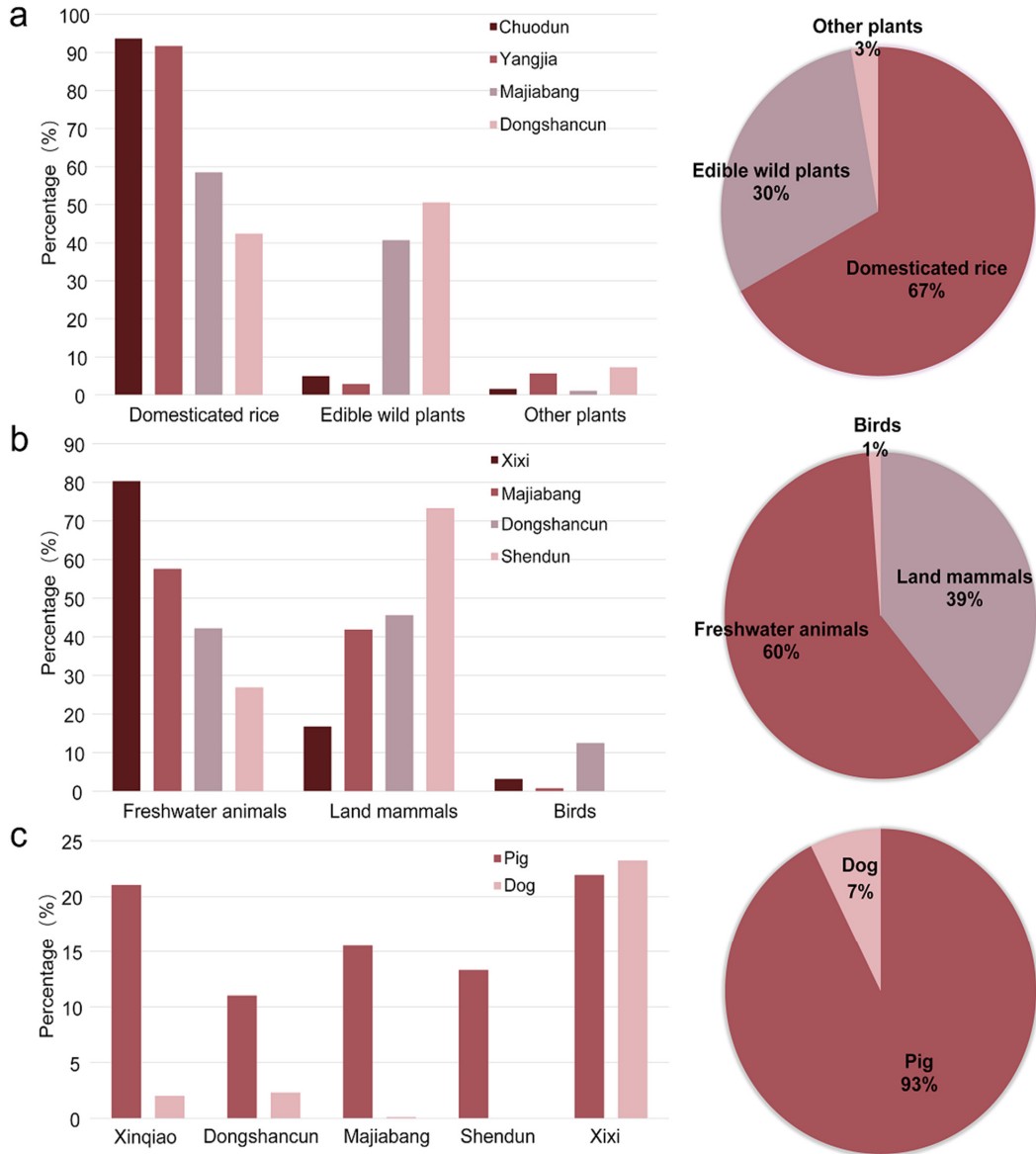

**Figure 3.** Comparison of different food resources during the Majiabang Cultural period. Percentages (**left**) and relative proportions (**right**) of the three plant groups from the four sites (**a**); percentages (**left**) and relative proportions (**right**) derived from the NISP data of the three animal groups from the four sites (**b**); percentages of pig and dog among mammals (**left**) and relative proportions (**right**) of pig and dog from the five sites (**c**). Note: one rice spikelet base was equal to one rice grain in the statistical analysis, and the same hereinafter.

The major vegetation type in the Taihu Lake area had been forest since the mid-Holocene. The edible wild plants were primarily $C_3$ plants, as $C_4$ plants such as Panicoideae and Eragrostoideae grasses seldom grew in this area due to its warm and wet climate. The mean $\delta^{13}C$ values of the $C_3$ and $C_4$ plants were $-26.5‰$ and $-12.5‰$, respectively, in the lower Yangtze river valley [4], which are supported by the $\delta^{13}C$ values of the Neolithic herbivores and omnivorous animals in this region, with an about 5‰ isotope enrichment from plant to bone collagen, and a 9~14‰ isotope enrichment from plant to bone bioapatite (Table S3). Thus, the $\delta^{13}C$ values of the residents in three of the Majiabang Culture sites (Sanxingcun, Xudun, and Jiangjiashan) ranged from $-20.05 \pm 0.21‰$ to $-20.3 \pm 0.22‰$, and the $\delta^{15}N$ values ranged from $7.6 \pm 3.59‰$ to $10.4 \pm 0.5‰$ (Table 1; Figure 2), confirming that the people of the Majiabang Culture fed on food dominated by $C_3$ plants as well as terrestrial and freshwater meat. The higher mean $\delta^{13}C$ value of the Majiabang site's residents ($-11.7 \pm 0.4‰$) (Table 1) is attributed to the test material of bone bioapatite, which also indicates a $C_3$-dominated plant diet.

4.2.2. Songze Cultural Period (5.8–5.3 ka BP; 12 Sites)

Compared to that of the Majiabang Culture, the Songze Culture's food diversity significantly decreased, which was especially highlighted by the diminished variety of wild animal and plant resources (Table S1(I) and Table 2). The people of the Songze Culture only consumed eight types of edible wild plants, including water caltrops, semen euryales, cucurbits, melons, persimmons, grapes, peaches, and dark plums, and only seven types of wild animals, including deer, Caprinae, buffaloes, birds, freshwater fishes, molluscs, and amphibians. The total NTAXA of the food types decreased sharply to 22, with the NTAXA of plants and animals each being 11 (Table 2). The Shannon–Wiener and Simpson diversity values for plants declined to 0.45 and 0.21, respectively, which were much lower than those of the Majiabang Culture (Table 3). Among the agricultural products consumed, there were three types of cereals: rice, common millet, and soybeans; although, common millet was only found in the Qingchengdun site in a very small amount (two grains) [67]. Wild food resources were consumed as well as domestic animals (pigs and dogs); although, the universalities of wild animals except for that of deer (50%) were relatively low (8–25%). Rice and pigs were the most common food resources, with universalities of 75% and 58%, respectively. The food types used per Songze Culture site were also very limited (mean of 4.4 types/site), and among the sites that had been studied with both archaeobotany and zooarchaeology methods, the largest number of food resources (nine types) was found in the Songze site, which although a small number, was the largest number of resource types found.

The quantitative data from the Xiaodouli and Qingchengdun sites (Figure 4) showed that domesticated rice overwhelmingly dominated the plant-based food resources consumed during that time, with the percentage, relative percentage, and ubiquity of being unearthed being above 94%, and the proportion of wild plant resources being very small. The $\delta^{13}C$ value found in the human bones at the Songze site was $-19.89 \pm 0.5‰$, confirming the major contribution of the $C_3$ plants—especially rice—to the diet of the Songze Culture peoples. The $\delta^{15}N$ value of $10.9 \pm 1.7‰$ indicated that they mainly fed on terrestrial or freshwater animal meat (Table 1; Figure 2), and research on the animal remains at the Songze site showed that the number of identifiable wild animal specimens accounted for 70% of the types of animals consumed (Figure 4), which also indicated that wild animals were the most important meat source during the Songze Cultural period.

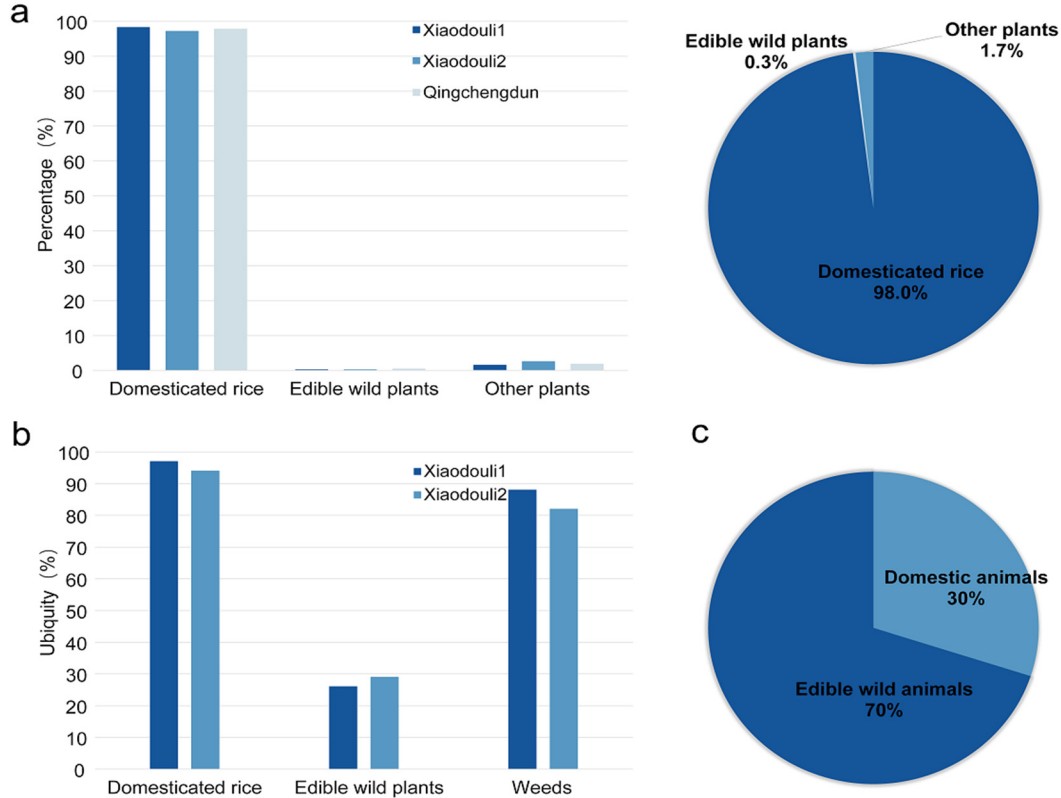

**Figure 4.** Comparison of different food resources during the Songze Cultural period. Percentages (**left**) and relative proportions (**right**) of the three plant groups from the two sites (**a**); Ubiquity of the three plant groups from the Xiaodouli site (**b**); relative proportions of domestic animals and edible wild animals at the Songze site (**c**). Xiaodouli1 and Xiaodouli2 in this figure indicate two sets of independent archaeobotanical data from the same site which can be seen in Table S1(II,IV).

4.2.3. Liangzhu Cultural Period (5.3–4.3 ka BP; 19 Sites)

In the Liangzhu Cultural period, people continued to consume three types of cereals (rice, foxtail millet, and soybeans) and two types of domestic animals (pigs and dogs). The diversity of wild food resources was in between that of the Majiabang and Songze Cultures (Table S1(I) and Table 2). There were 14 types of edible wild plants (lotus seeds and hawthorns were not found) and 9 types of wild animals (deer, buffaloes, boars, birds, freshwater fishes, molluscs, amphibians, marine fishes, and shellfishes). The number of food resources consumed by people in the Liangzhu Culture accounted for 74% of the available total food types (Table S1(I)). The total NTAXA of food types increased to 44, with the NTAXA of plants and animals being 17 and 27, respectively (Table 2). The Shannon–Wiener and Simpson diversity values of the plants were similar to those from the Songze Cultural period, and those of animals were much lower (0.92 and 0.34, respectively) than those from the Majiabang Cultural period (Table 3). Among the agricultural products, rice was the most common food resource, with a universality of 95%, whereas domestic pigs and dogs had universalities of 47% and 32%, respectively. Foxtail millet and soybeans were only unearthed at the Bianjiashan and Chuodun sites, respectively. Other common wild resources and their universalities were melons (53%), semen euryales (47%), cucurbits (47%), jujubes (47%), peaches (47%), water caltrops (32%), deer (37%), buffaloes (37%), and freshwater fishes (37%). The number of food resources used per site was high (mean of 7.7 types/site). The Liangzhu Ancient City ruins had the greatest variety of food resources, among which the Bianjiashan and Meirendi sites had 21 and 18 varieties of food, respectively.

The quantitative data from the five sites (Figure 5a) showed that domesticated rice remained the major plant-based food consumed during the Liangzhu Cultural period

(33–94%; relative percentage of 86%), and that the proportion of wild plants consumed was relatively small (3.2–51.3%; relative percentage of 7%). The major meat source was terrestrial mammals (50–100%; relative percentage of 98%), supplemented by freshwater animals (0–50%; relative percentage of 1%) (Figure 5b). Among mammals, domestic pigs accounted for the highest proportion of food consumed (21–77%; relative percentage of 91%) (Figure 5c). Isotopic evidence from the Zhuangqiaofen and Meirendi sites (Table 1; Figure 2) also showed that the major food sources of Liangzhu residents were C$_3$ plants and terrestrial mammals. In the residue analysis of Xiawan site, trace amounts of fatty acids from marine animals, freshwater animals, and wild ruminants were found, and the proteomics research found food components such as rice, large yellow croakers, and freshwater fishes from the carbonised materials [55]. The fatty acids of marine organisms found at the Xiawan site [58] showed that seafood was a part of the diets of some Liangzhu people.

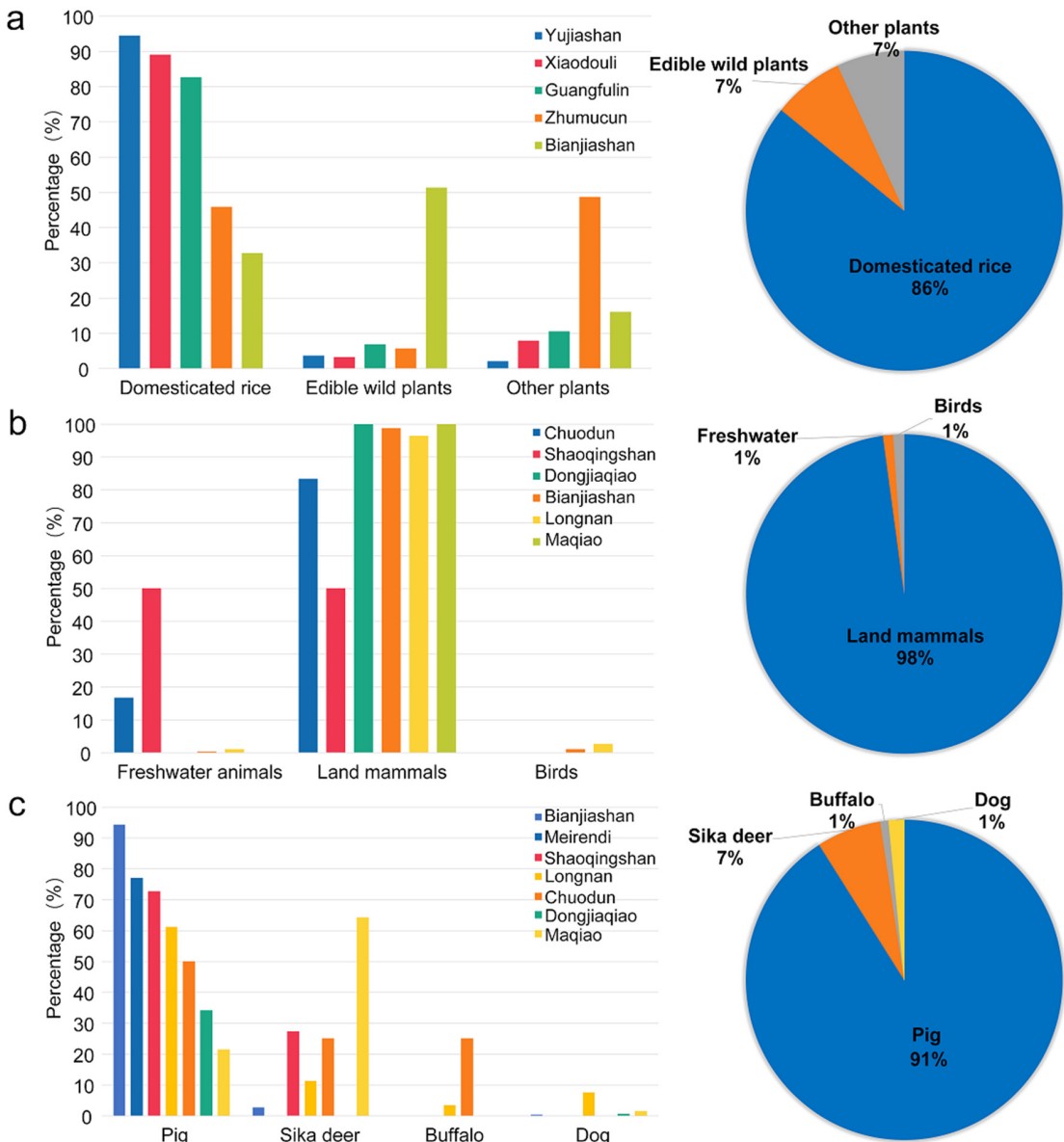

**Figure 5.** Comparison of different food resources during the Liangzhu Cultural period. Percentages (**left**) and relative proportions (**right**) of the three plant groups from the five sites (**a**); percentages (**left**) and relative proportions (**right**) derived from the NISP data of the three animal groups from the six sites (**b**); percentages (**left**) and relative proportions (**right**) derived from the NISP data of pig, dog, sika deer, and buffalo from the seven sites (**c**).

4.2.4. Temporal Changes in Diet Structures

The temporal changes in agricultural products as part of the peoples' diet structure were examined by calculations based on the published archaeobotanical and zooarchaeological quantitative data: the percentage of cereals (rice and millets) from the total number of unearthed seeds, and the percentage of the identifiable specimens of domestic animals from the total number of animals (Table S1(II,III)). Figure 6 shows that the proportion of crops amongst the plant-based foods increased from 67% during the Majiabang Cultural period to 98% during the Songze Cultural period, then decreased slightly (to 86%) during the Liangzhu Cultural period. These percentages show that crops had always been the staple plant-based food in the Taihu Lake area during the Neolithic era. The proportion of domestic animals as a food source in the Majiabang Cultural period was extremely low (7%), but it increased rapidly to 30% during the Songze Cultural period, then increased significantly to 82% during the Liangzhu Cultural period. Therefore, it was only during the Liangzhu Cultural period that domestic animals became the main meat source for the Neolithic people.

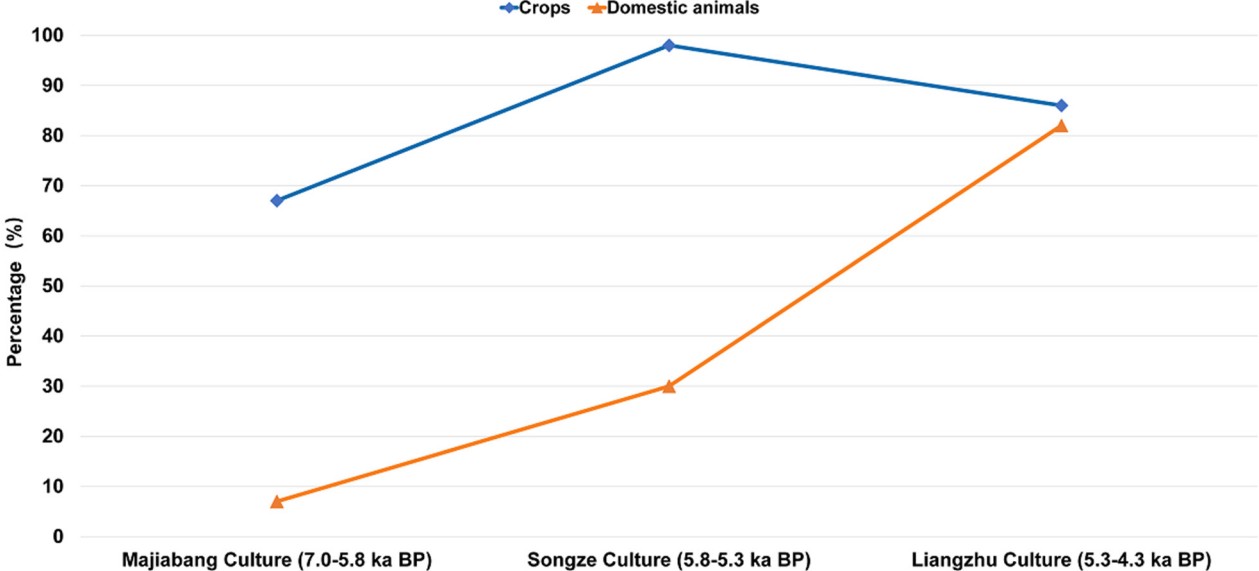

**Figure 6.** Temporal changes in the relative proportions of crops and domestic animals consumed by the Neolithic people in the area around Taihu Lake.

The NTAXA of edible wild plants changed slightly from the Majiabang Cultural period (15) to the Songze Cultural period (8) and the Liangzhu Cultural period (14). In contrast, the NTAXA of wild animals drastically declined from the Majiabang Culture (49) to the Songze Culture (9) and to the Liangzhu Cultural period (25), which was especially highlighted by the NTAXA of freshwater animals and land mammals (Table 2). This implies that the diversity of wild animals in the human diet was also reduced along with their declining proportions. The diversity values for wild plant-based and animal-based foods showed a parallel pattern (Table 3; Figure 7), in which both decreased from the Majiabang Culture to the Songze and to the Liangzhu Cultural period. This could suggest that crop and livestock agricultural development is associated with the declining diversity of wild food resources in diet structures.

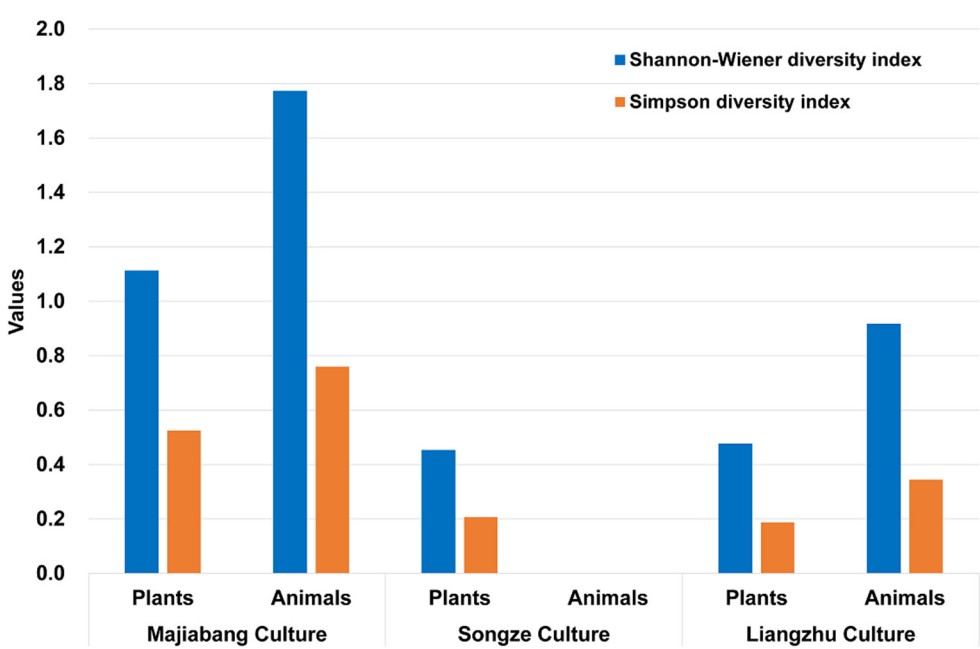

**Figure 7.** Bar chart of Shannon–Wiener and Simpson diversity values for Neolithic plant and animal foods for each period in the area around Taihu Lake.

## 5. Discussion

*5.1. Subsistence Economy in the Area around Taihu Lake during the Neolithic Era and Its Relationships with the Environment and Society*

Due to the suitable climate, vegetation, and hydrological conditions during the mid-Holocene, the Neolithic people living around Taihu Lake consumed a diet that included a wide variety of food, providing the preconditions for the development of various types of subsistence economies. Across the different periods, regardless of whether rice agriculture and animal husbandry were the main economic activities, people continued to exploit wild animal and plant resources by gathering, fishing, and hunting. The positive relationship between plants and animals in their temporal changes of diversity (Figure 7) indicates that people may have exploited plant- and animal-based foods from the same environmental zones. The freshwater wetlands and plains that gradually formed around the Taihu Lake from 7 ka BP onwards determined the subsistence pattern of humans—one that was based on utilising wetland resources. Wetland plants such as paddy rice, water caltrops, and semen euryales were the main sources of starch, and freshwater fishes, molluscs, and turtles (NTAXA = 30) provided high-quality protein. Terrestrial meat resources (NTAXA = 15) such as deer, buffaloes, and birds (NTAXA = 7) also inhabited wetland environments. The paddy field cultivation system that developed throughout the Neolithic era was closely related to the exploitation of marshlands [68].

In the lower reaches of the Yangtze River, where the Taihu Lake area is located, people commenced rice production and animal husbandry around 10–7 ka BP. While residents in certain regions of the area fished in the ocean, hunting and gathering still dominated the subsistence economy [43]. Such evidence mainly comes from some early sites, such as the Shangshan site clusters (i.e., the Shangshan, Hehuashan, Qiaotou, and Huxi sites) [69–72], and the Xiaohuangshan [73,74], Jingtoushan [75–77], and Kuahuqiao sites [78–80], which are all located south of the Qiantang River. The sites were all located within a combination of mountains, hills, and river valley plains, which protected them from the impact of the drastically fluctuating sea levels at that time. However, the Kuahuqiao site adjacent to the Taihu Lake area and the Jingtoushan site in the Yaojiang River Valley were eventually destroyed by transgressions, which interrupted the development of agriculture [77,78]. With the marine regression after 7 ka BP and the gradual formation of the Yangtze River Delta, land areas (terraces and slopes) suitable for human settlement, wetlands, and freshwater

resources appeared on the edges of the saucer-shaped depressions centred around Taihu Lake [20,32,36,81]. Combined with the warm and humid climate, the terrestrial ecology rapidly diversified, and abundant plants, animals, fishes, and shellfishes flourished. This increase in flora and fauna diversity and the availability of land provided wide spaces and subsistence resources for human production and the development of livelihoods. In such an environment, the Neolithic Culture that originated from the Majiabang Culture around the Taihu Lake developed and thrived.

### 5.1.1. The Majiabang Culture

Abundant remains of rice (grains and rachilla) were found in the Majiabang Culture's sites. In their Majiabang, Caoxieshan, Chuodun, Jiangli, and Chenghu sites, larger-scale rice paddy fields with supporting facilities such as water storage pits, ditches, ponds, and wells were found. These facilities were connected to form a complete circular irrigation system, demonstrating that paddy field cultivation was already relatively mature during the Majiabang Cultural period [51,82–86]. The rice grain sizes increased, and the morphologically domestic rachilla proportions of 70% or even >95% [82,85–88] far exceeded the proportions of domesticated rice in the Yangtze and Huai river basins (Shunshanji and Hanjing) in the early stages (9–7 ka BP) [70,89,90]. This indicates that rice domestication culminated in the late Majiabang Culture (6.5–5.8 ka BP), and that rice was their major plant-based food, as demonstrated by its universality and percentage, despite the results of this study showing that the Majiabang people used a broad-spectrum feeding strategy and consumed a diet that contained the largest variety of food (Figure 3). Fruit plants such as water caltrops, semen euryales, cucurbits, melons, peaches, and dark plums were only supplementary food sources. This comparison demonstrates that the advancement of rice domestication, the establishment of its rice production system, and the increase in yields allowed paddy rice cultivation to become the main component of the Majiabang Culture's subsistence economy. However, due to the sparse sites and data from the early Majiabang Cultural period (7–6.5 ka BP), it is difficult to determine whether the Majiabang Culture had transitioned from a gathering subsistence economy to rice agriculture or if it had relied heavily on rice cultivation since the beginning of their culture's establishment.

The Majiabang Culture obtained meat resources mainly by catching freshwater animals and hunting wild mammals. Domestic pigs and dogs accounted for a tiny proportion of the meat supply, out of which domestic pigs were the more important source. It should be noted that although dogs are regarded as a source of meat in this study, it is unlikely that the dogs were intentionally raised for meat, as represented by their low consumption proportion percentages (primarily < 5%). Other functions such as hunting, companionship, guarding, or acting as pets were probably more important than supplying meat. However, at the Xixi site, where the consumption proportion of dogs was higher than that of pigs (Figure 3c), dogs were possibly raised for meat. However, raising dogs for meat in the Neolithic Taihu Lake area was generally restricted. Due to the favourable aquatic environment around the sites, aquatic fishes, molluscs, and turtles (NTAXA = 25) were meat resources that were more accessible to people compared to wild terrestrial animals (NTAXA = 13) such as deer and buffaloes [91,92]. The various types of bone arrows and weights attached to a fishnet found at the site demonstrate that the Majiabang people residing in this area were proficient in fishing [93]. There were even pits for breeding freshwater fishes and turtles in the Luojiajiao and Majiabang sites, which would have enabled consistent and large-scale fishing [93]. Stacks of fish, shellfish, and snails up to 0.3–1.2 m were found in the cultural layers of the Majiabang, Sanxingcun, and Xudun sites, which further show that a huge quantity of aquatic animals was caught and consumed during the Majiabang Cultural period [93]. While such enormous terrestrial and freshwater resources in the wetland discouraged people living around the Taihu Lake area from developing their use of marine resources [68], various marine fishes and whales (NTAXA = 5) were lightly fished in a few sites (Dongshancun, Luojiajiao and Songze).

The Lixiahe and Ningshao plains are adjacent to the area around Taihu Lake (Figure 1) and supported the Longqiuzhuang and Hemudu cultures, respectively, at the same time the Majiabang Culture existed. The three cultures had similar environments, resources, and subsistence means; they developed rice agriculture, gathered water caltrops and semen euryales as supplemental food, and used terrestrial and freshwater wild animals as the main meat sources (Tables S1(I) and S2(I)). Moreover, the people of all three cultures seldom used marine resources (except for at the Hemudu site) [43,94]. The differences between the three cultures were as follows:

(1) In terms of the degree of rice domestication, the planning and management of the paddy fields, as well as the proportion of rice in the diets and for subsistence, the Majiabang Culture's rice agriculture was the most advanced, which is related to the stable environment of freshwater wetlands around the Taihu Lake that had been maintained for thousands of years. In contrast, the Lixiahe Plain (where the Longqiuzhuang site is located) had been an area of coastal lagoons before 6.0–5.5 ka BP [32], and the Ningshao Plain (where the Hemudu Culture site is located) was largely influenced by the sea-level fluctuations during the same time. Although rice cultivation and domestication started in the Ningshao Plain as early as 7–6.7 ka BP [95], the marine transgressions during 6.4–5.6 ka BP induced frequent transformations of the environment from freshwater wetlands to intertidal mudflats, thus interrupting the process of rice cultivation. Therefore, rice domestication in the region was not consistent and established until 5.6 ka BP [96–98].

(2) Compared to the Majiabang and Longqiuzhuang cultures, the Hemudu Culture began consuming acorns early. This is attributed to the fact that the Hemudu Culture sites (located in the Yaojiang Valley) were close to forest edges [99], while the Majiabang and Longqiuzhuang culture sites tended to be on gentle slopes with open spaces or terraces, further from forest edges and their resources. Given that a considerable amount of labour was required for acorn gathering and rice cultivation alike, there was no means to achieve both at the same time. The Majiabang people residing in this area chose to work intensively on rice agriculture, whereas those of the early Hemudu Culture had an insufficient rice supply and thus had no choice but to gather and store acorns for survival [100]. However, from 6.7 ka BP, the Hemudu Culture drastically reduced the gathering of acorns, focusing instead on utilising water caltrops, semen euryales, and rice. From 6.0 ka BP onwards, they even began to clear forests for rice agriculture [101].

In summary, in a warm and humid climate with freshwater wetlands, rice agriculture along with fishing and hunting became the Majiabang Culture's primary means of subsistence and maintenance of their food supplies. The $\delta^{13}$C and $\delta^{15}$N values of human bones corroborate that the Majiabang people in the region had a relatively balanced diet of plants and meat (Table 1; Figure 2), which had the highest food richness of any diet among the Neolithic Taihu Lake area cultures. Such a subsistence economy that was fully adapted to utilising the wetland ecological environment laid the foundation for the Majiabang people to establish settlements relying on wetlands and develop jade ware production and techniques to build large wooden constructions. These factors allowed them to create more wealth, which in turn allowed social stratification to occur [93].

### 5.1.2. The Songze Culture

There has been controversy over whether hunting-gathering or agriculture was the main subsistence economy of the Songze Culture [67,82,102]. While this study showed that (Figure 4) compared to those of other plant resources, the universality, percentage, and ubiquity of domesticated rice were overwhelmingly high, it is worth noting that the current result was derived from quantitative archaeobotanical data of only two sites. Thus, more data are needed to examine whether these high utilisation values of domesticated rice are widespread in Songze Culture sites. However, the number of wild plant types gathered was significantly lower than that of the Majiabang Culture, and the universalities and proportions had decreased to an extremely low level. Although raising domestic pigs had become more common, the dietary proportion of pork remained far below that

of wild animals (deer were the most common). Therefore, the evidence suggests that a subsistence economy primarily consisting of rice agriculture, hunting, and fishing was firmly established in the Songze Cultural period, whereas gathering wild plants and husbandry merely provided supplementary subsistence.

The temperature and precipitation during the Songze Cultural period decreased from the previous period, though the environment remained warm and wet [37]. The continuous retreat of the coastline led to the exposure of more terrestrial or freshwater resources such as saucer-shaped depressions, estuaries, plain depressions, and marshlands. This favourable hydrology, topography, and soil conditions allowed for the migration and settlement of people and the rapid development of rice agriculture [20]. The archaeological records show that the number of Songze Culture archaeological sites is approximately twice that of the previous period, indicating a surge in population and a deepened degree of settlement [42,67]. This increased population enabled the residents to devote more energy to developing paddy rice production, improving rice field farming techniques, and producing specialised agricultural tools, thereby ensuring the stability and reliability of food resources through rice cultivation. By using a stable and high-yield food source such as rice as the dominant food for subsistence, the Songze people also adopted a strategy to adapt to populational and environmental changes. Moreover, in the tombs of the Dongshancun site, the rich and poor were buried separately, providing evidence that the Songze Culture was advancing into an agricultural society and showing the first signs of social stratification in Neolithic China. This indicates the period in which the social complexity process in the Taihu Lake area began [43].

### 5.1.3. The Liangzhu Culture

The Liangzhu Culture (5.3–4.3 ka BP) has been regarded as the symbol of the establishment of a rice agriculture society and the beginning of Chinese civilisation [100,103]. The subsistence economy was mainly supported by intensive rice production and the raising of domestic pigs [21]. The results of this study support this point of view. Rice accounted for 92% of the edible plant resources, and domestic pigs accounted for 91% of the major meat resources from mammals (Figure 5). Compared to the previous two periods, freshwater animals during the Liangzhu Cultural period accounted for a significantly lower proportion of food in the people's diets (only 1%), with their NTAXA declining to 13. This demonstrates that the importance of freshwater fishing for supplying protein had decreased and had been replaced by the breeding and hunting of terrestrial animals (NTAXA = 5; mainly deer). The varieties and proportions of edible wild plants increased from those of the Songze Cultural period, and some of the varieties (including water caltrops, semen euryales, cucurbits, melons, jujubes, and peaches) may have been planted as horticultural plants. This is especially true for fruit trees such as peach trees, which had obviously been domesticated [104].

Yuan et al. [105] pointed out that although rice cultivation and the raising of domestic pigs were common in the Liangzhu Culture, the subsistence economy of the Liangzhu Culture was extremely unbalanced across different locations. The Liangzhu Ancient City and its surrounding area in Yuhang was the centre of Liangzhu Culture [46]. The discovery of paddy fields, complex irrigation facilities (Maoshan site), relics of fertilization [54], and 13,000 kg of rice in the Mojiaoshan granary [106] demonstrate the existence of prosperous rice agriculture. Within the sites around the Taihu Lake, the proportions of domestic animals were mainly >50%, but in sharp contrast could be as high as 90% in the central area of Liangzhu Culture or far lower in several sites away from the centre (such as the Guangfulin and Maqiao sites; Figure 5) [105]. The Lixiahe Plain in the eastern Jianghuai area and the Ningshao Plain along the coast of eastern Zhejiang were at the edge of the Liangzhu Culture region. These locations saw the formation of stable freshwater wetland environments after 5.5 ka BP [32,107] and whose inhabitants had mastered the techniques for rice paddy farming and raising domestic pigs before the Liangzhu Cultural period. However, the zooarchaeological, archaeobotanical, and isotopic evidence from specific sites

in these regions—Jiangzhuang, Kaizhuang, Qingdun (eastern Jianghuai) [108–110], Cihu and Tashan (coast of eastern Zhejiang) [61,105]—all show that the Liangzhu residents used gathering, fishing, and hunting as the main subsistence. This further shows the drastic modal and social organisational differences regarding agricultural production between these outer regions and the central Liangzhu Culture area.

During the Liangzhu Cultural period, the climate around the Taihu Lake tended to be cool and dry, and the vegetation transformed into a forest-steppe. However, the wetland ecosystem remained stable [111]. The wide range of environments was favourable for rice cultivation and the exploitation of the abundant wild animal and plant resources. Therefore, while the Liangzhu Culture did not develop a subsistence economy balanced across all regions, they did—in appropriate environments and under the influence of a long-standing agricultural tradition—develop an advanced and stable agricultural economy, which is where the production of foods, including rice and pork, reached a historic high. Meanwhile, the number of settlements belonging to the Liangzhu Culture increased rapidly in the Taihu Lake area. As a result, population growth, urbanisation, and a bureaucratic system separate from agricultural production emerged [47]. Based on the symbolic jade objects (*cong* and *bi*), and jade artefacts whose decorations are dominated by recurring man/beast motifs, it is clear that the Liangzhu culture had a unified belief in a system of deities and spirits [47]. This formation of theocratic power, royal power, and a political system marked the birth of an early state and civilisation in Liangzhu [46]. Their solid economic foundation and spiritual strength led to strong cohesion within their society, which allowed it to persist through damage caused by floods, typhoons, storm surges, and other natural disasters throughout a 1000-year period [105]. However, in 4.4 ka BP, saltwater from the rapid rise in the relative sea level intruded into the Taihu Lake area, depositing a yellow silty soil over a large area that destroyed the wetland ecosystem so crucial to rice agriculture [40]. Furthermore, large-scale floods due to the abnormal climate that were followed by long-term drought caused a sharp decrease in wild resources [41]. Subsequently, and perhaps consequently, the subsistence economy, which had relied on the cultivation of a single crop (rice) and the husbandry of a single species (pig), collapsed expeditiously, leading to the decline of the Liangzhu Culture.

*5.2. An Exotic Crop: Millets*

A few remains of millets were found in the Taihu Lake area, including nine grains of foxtail millet unearthed from the Yangjia site of the Majiabang Culture, two grains of common millet unearthed from the Qingchengdun site of the Songze Culture, and one grain of foxtail millet unearthed from Bianjiashan site of the Liangzhu Culture (Table S1(I,II)), none of which have been directly dated by radiocarbon dating. The available evidence is insufficient to indicate that millets had been grown in the Taihu Lake area, which during the Neolithic era had abundant precipitation, wetland environments, and sticky soil—all of which are not favourable for dryland crops. Moreover, prosperous rice production and abundant wild resources were sufficient to ensure a food supply, i.e., introducing dryland crops was unnecessary. Therefore, the millets were more likely to have come from the exchange of grains with northern areas (i.e., the Haidai region or the Huai River basins), as there is no evidence of dryland farming in the Taihu Lake area.

Given that from 5.5–5.0 ka BP millet remains have been discovered in the coastal areas of Fujian, Taiwan, and Guangdong [112–115], scholars have suggested two possible routes along which dryland crops may have been brought to the southeast during the Neolithic era: (1) the crop seeds travelled by sea from Shandong to the southeast coast, or (2) the crops were brought from southern Hubei, northern Hunan (i.e., Chengtoushan site), and Anhui to Fujian and Guangdong via Jiangxi [115–118]. In the future, if millet grains from the Neolithic era with direct dating evidence are found in the Taihu Lake and Zhejiang area, the discovery of a new route of the southward spread of millet farming (being spread on land from the Haidai region to the Taihu Lake area and Zhejiang via eastern Jianghuai, and finally reaching the southeast) is likely. A recent discovery in the Anle site (i.e., Yaodun site

in Figure 1) of the Songze Culture in northern Zhejiang of the earliest foxtail and common millet dating to earlier than 5750 BP may provide vital clues for this supposed route [119].

## 6. Conclusions

By applying an analysis from the perspective of food resources, this paper discusses the relationships between the subsistence economy, environmental changes, and social evolution in the Taihu Lake area during the Neolithic era. The suitable climate and stable environment of freshwater wetlands provided the Neolithic residents with abundant food resources (approximately 38 varieties of edible animals and plants). Paddy rice cultivation, which developed in wetlands, has always dominated the subsistence economy. An agricultural society, whose economy was dominated by rice production and to whom the process of social complexity occurred, emerged in the Songze Cultural period. Finally, early civilisation developed in the Liangzhu Cultural period.

Compared to the prosperous rice agriculture, the development of husbandry in the area around Taihu Lake during the Neolithic era was lacking. It would have relied heavily on raising domestic pigs, and during the 2000 years of the Majiabang and Songze Cultural periods it was not able to provide sufficient meat. During this time, the fishing of freshwater animals and the hunting of wild mammals were the main sources of meat. It was not until the Liangzhu Cultural period that pig farming became the main meat source. Such an unbalanced development of crop cultivation and husbandry was rare in the ancient world. Arguably, the Taihu Lake area experienced a complex society and early civilisation that had only wetland ecological resources and an economy based on paddy rice. This unique evolutionary path of societal prosperity and collapse has important implications for present-day humans with respect to constructing and ensuring an ecological civilisation based on resource sustainability.

**Supplementary Materials:** The following supporting information can be downloaded at: https://www.mdpi.com/article/10.3390/land11081229/s1. Table S1: Qualitative and quantitative data from archaeobotanical and zooarchaeological research on the area around Taihu Lake during the Neolithic era. Table S2: Qualitative data from archaeobotanical and zooarchaeological research on the Lixiahe Plain and Ningshao Plain during the Neolithic era. Table S3: $\delta^{15}$N and $\delta^{13}$C values of the animal bones from the five Neolithic sites in the lower reaches of the Yangtze River. References [120–195] are cited in the supplementary materials.

**Author Contributions:** Conceptualization, C.W. and Y.G.; methodology, C.W., Y.G., Y.W. and Z.Z.; formal analysis, C.W. and Y.W.; data curation, Y.W.; writing—original draft preparation, Y.W. and C.W.; writing—review and editing, C.W. and Y.G. All authors have read and agreed to the published version of the manuscript.

**Funding:** This work was supported by the Young Taishan Scholars Program (Grant No. tsqn201909009), the National Natural Science Foundation of China (Grant No. 42072032), the Shandong University Multidisciplinary Research and Innovation Team of Young Scholars (Grant No. 2020QNQT018), the Program of Research on Education and Teaching Reform of Shandong University (Grant No. 2022Y057) and the National Key Research and Development Program (Grant No. 2020YFC1521606).

**Institutional Review Board Statement:** Not applicable.

**Informed Consent Statement:** Not applicable.

**Data Availability Statement:** The original contributions presented in the study are included in the article/Supplementary Materials. Further inquiries can be directed to the corresponding authors.

**Acknowledgments:** We thank Yu Gao and Haiming Li for providing their valuable archaeobotanical data, and Yajie Dong for his kind help in the data processing. We also appreciate the constructive comments and suggestions from the editor and two reviewers.

**Conflicts of Interest:** The authors declare no conflict of interest.

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
