# Peer review of "Subsistence, Environment, and Society in the Taihu Lake Area during the Neolithic Era from a Dietary Perspective"

_land, doi:10.3390/land11081229_

Round 1

Reviewer 1 Report

The Article "Subsistence, Environment and Society in the Area around Taihu Lake during the Neolithic: Perspectives from Diet on Animals and Plants" proposes an interesting overview and thorough review of neolithic subsistence in an extensive area around Taihu Lake.  Even though the analyses of the temporal variation of faunal and plants/cereal consumption showed changes through the three cultural periods analyzed,  I will point to several issues that I think should be addressed to get a real notion of the dimension of such variability.

-The analyzed area is very extensive, therefore, when talking about the sites by period, it is necessary to know where those sites are located. The variability in the exploited resources may be due not only to cultural differences but also to the variation in the availability of resources by area.  Figure 1 shows sites near the sea and sites in Highlands more than 120 km from the coasts. 

-Isotopic analyses should include information about the isotopic ecology of the area. For example, which are the C3 and C4 plants? Which is the isotopic values of the herbivores? of the consumed animals?

-Faunal and Botanic analyses should have a better statistic analysis. I encouraged the authors to use the most common index that zooarchaeological and archaeobotanical studies used such as NTAXA (to measure taxonomic richness in each period) and the Simpson diversity index.

See:

GRAYSON, Donald K. Quantitative zooarchaeology: topics in the analysis of archaeological faunas. Elsevier, 2014.

LYMAN, R. Lee. Quantitative paleozoology. Cambridge University Press, 2008.

VANDERWARKER A.M. 2010 Simple Measures for Integrating Plant and Animal Remains. In: VanDerwarker A., Peres T. (eds) Integrating Zooarchaeology and Paleoethnobotany. Springer, New York, NY

 Comments on Tables and Figures:   

In figure 1, I think it is important for this paper to differentiate the cultural period of each site. The geographical location of each site may probably condition the exploited resources. For example, site nº 19 is close to the ocean, site 42 is inland and at a high altitude. Resource availability should not the same in both areas

Table 1S is an excel file with a lot of tables. You should be clearer with the materials you used for the analyses. This table is remitted in many parts of the manuscript but the reader does not know which table should look at.

I also made several comments in the manuscript which I think the authors should address. 

I hope my comments help to improve the  manuscript

Reviewer 2 Report

This paper summarized data concerning subsistence activities at Neolithic sites located around Taihu Lake. Quantitative data on faunal remains and plant remains were taken from published reports and the presence or absence of taxa as well as their relative proportions were compared to see the changes through time and space. The temporal changes and spatial difference in the exploitation of food resources are discussed. Results of carbon and Nitrogen isotope analyses from the Neolithic sites were also summarized.

The major contribution of this paper would be the amount of data presentation that would serve as a reference for archaeologists in the investigation of development of rice paddy agriculture and resource fluctuation that might have been caused by environmental changes.

There are a few points that the present reviewer sees as problematic, although they do not seriously affect the course of discussion of the paper.

1) Materials and methods section needs more description on the analysis.

- Page 5 Line 1: Please explain the kind of "residue analysis" and the method. (Analysis of fatty acid?)

- Table S1 and S2 contain five and two tables respectively. Each table should be numbered separately and referred in the text.

- Page 5 second paragraph: The explanation of the data sets is not easy to follow.

Please explain "system sampling", "systematic research data", "residue data"

Please explain "kinds and universality of food resources". Do the authors mean the variety or the range of food resources by "kinds"? By universality, I assume the authors mean the proportion of the number of the site where certain taxon was found. More explanation is necessary in the method section.

Please explain "ubiquity". Is it the same as "universality"?

I assume that the "percentage" was calculated for each site to show the proportion of the number of a taxa in the total counts of identified taxa, but please explain.

Page 10, second paragraph "relative percentage of 91 %" Please explain how the relative percentage was calculated. How does it exceed the highest proportion of 21-77%?

2) Dogs are classified as one of the animals kept for meat supply and discussed together with pigs. This is probably not correct. Unlike pigs, dogs have multiple functions depending on the cultural background of the people who keep the animal. Dogs could be kept as hunting companion, especially in the places where wild pigs were actively hunted, or pets, or for meat. It may also be the case that some dogs were kept for meat and some as pets at the same site. When discussing dogs as a source of meat, presence of cutmarks and representation of skeletal elements should be examined carefully. At sites where the proportion of dogs was higher than pigs (such as Xixi in Figure 3), it is possible that they were kept for meat. It is unlikely that the dogs were raised for meat, however, at sites where the proportion of dogs was less than 5 percent.

3) P.17, last sentence:

"an ecological civilization" please explain what is implied by "ecological". Do the authors mean sustainability of the resources?

Other minor comments

There are some words or expressions that are not suitable.

"freshwater animals" should be changed to "freshwater food resources" or "freshwater fish, mollusks, amphibians, and turtles".

The word "ancestors" often appears in the text. This word should be replaced by something like "the residents of the site".

P.2, 2.1

"east of the East China Sea" should be "west"

P. 13, 5.1.1. Line 7.

domesticated rachilla → morphologically domestic type rachilla

Line 8 "the domestication degree of rice" Please explain.

P.13, Line 11  from the bottom: Stacks of

Was the deposit found in a pit or in a midden?

P.15, second paragraph,

 Line 3: "warm and humid warm and dry" Please clarify.

 Line 7: "relic sites" meaning "archaeological sites"?

P.15, 5.1.3, Line 9 "meat supply" protein supply

P.16, second paragraph, Line 15

"strong spiritual strength led to strong cohesion" please paraphrase.

P.17 millets

Carbonized grains of foxtail millets were reported in the Daxi 大溪Culture period contexts (5800 BP) at Chengtoushanin Hunan (Nasu et al. 2007; 2012). This finds should be cited, as it can be a supporting data for the route 2).

Nasu, H., A. Momohara, Y. Yasuda, and J. He. 2007. The occurrence and identification of Setaria italica (L.) P. Beauv. (foxtail millet) grains from the Chengtoushan site (ca. 5800 cal B.P.) in central China, with reference to the domestication center in Asia. Vegetation History and Archaeobotany. 16:481-494.

Nasu, H., H-B. Gu, A. Momohara, and Y. Yasuda. 2012. Land-use change for rice and foxtail millet cultivation in the Chengtoushan site, central China, reconstructed from weed seed assemblages. Archaeological and Anthropological Sciences. 4: 1-14.

Figures

The fonts in Figure 3 and 4 should be larger. It is hard to read.

Round 2

Reviewer 1 Report

The authors made substantial changes to the manuscript that contributed to its improvement. All the suggestions I made in the first evaluation were taken into account. I am glad to have collaborated with you in this work through my evaluation.

English improved noticeably, although there may be some minor issues that are beyond my knowledge of the language.

I consider that the manuscript is in condition to be published in Land.